# Effect of Biodiesel on Performance of Cold Patch Asphalt Mixtures

**DOI:** 10.3390/ma17225566

**Published:** 2024-11-14

**Authors:** Lingchen Bao, Rongxin Guo, Feng Yan

**Affiliations:** School of Architecture and Engineering, Kunming University of Science and Technology, Chenggong District, Kunming 650500, China; 20222110011@stu.kust.edu.cn (L.B.); guorx@kmust.edu.cn (R.G.)

**Keywords:** road engineering, biodiesel, cold patch asphalt, FTIR, adhesion, road performance

## Abstract

In order to reduce the amount of diluent in a diluted asphalt mixture, this study developed a cold patch asphalt (CPA) for repairing pavement potholes by using a mixture of treated biodiesel and diesel as the diluent. The effects of biodiesel on the performance of the cold patch asphalt mixture (CPAM) during the construction process were investigated through Brookfield rotational viscosity tests, adhesion tests, and FTIR (Fourier transform infrared spectroscopy) analyses. At the same time, the effect of biodiesel on the performance of the CPAM was analyzed by combining the strength growth test, rutting test, and water-soaked Marshall test of CPAMs. The test results show that the construction performance of the CPAM can be significantly improved by adding pretreated biodiesel. Under the same amount of diluent, the strength and high-temperature performance of the asphalt mixture diluted with biodiesel were significantly improved compared to that with diesel as the diluent. The optimal high-temperature performance reached 9027 (times/mm), representing an approximate increase of 94.7% compared to 4636 (times/mm) when only diesel was used as the diluent. When the biodiesel content increased from 10% to 40%, the residue stability improved from 85.9% to 91.3%. The corresponding 0.5 h Marshall stability increased from 5.59 kN to 8.1 kN, while the 48 h Marshall stability rose from 4.8 kN to 7.39 kN. All tests met the requirements for hot mix asphalt.

## 1. Introduction

A CPAM is produced by blending emulsified, diluted, or foamed asphalt with unheated aggregates. CPAM technologies hold significant economic, ecological, and production advantages over hot mix asphalt (HMA) and are becoming a focus of research as a pavement material [1,2,3,4]. However, CPAMs also have some shortcomings, including a high porosity, weak early strength, and a considerably long curing time to reach full strength [5,6,7]. Traditional cold mix asphalt typically uses kerosene and diesel as diluents, with large amounts of these solvents added to achieve the required fluidity for construction. However, the heavy use of these solvents can negatively impact the environment and also lead to poor performance and high costs of the cold patch materials. To enhance its performance, additives are often incorporated into cold patch asphalt (CPA). These additives mainly include solvents that facilitate dissolution, rubbers and resins that enhance performance, and surfactants that improve adhesiveness [8].

Biodiesel is an ester-based oxygen-rich renewable fuel, primarily produced from various raw materials such as algae, animal fats, microbial oils, and vegetable oils (soybean, rapeseed, and palm oils) [9]. Biodiesel shares many characteristics with conventional diesel and has notable properties including high biodegradability and lubricity [10]. Due to its high flash point and biodegradable nature, biodiesel degrades 3–4 times faster than petroleum diesel, making its handling, transport, and storage safer [11]. Biodiesel exhibits superior lubrication properties and as reported by Ejaz et al. [12], it can be blended with diesel in specific proportions to obtain biodiesel mixtures. Zargar et al. [13] and Sun et al. [14,15] used various rheological tests to explore the effects of waste cooking oil (WCO) and its residues from biodiesel production on aged asphalt. Additionally, Sun et al. [16] found that derivatives of bio-oil prepared from WCO could reduce the complex modulus and creep stiffness while increasing the asphalt’s m-value and phase angle, enhancing its thermal crack resistance. According to a Fourier transform infrared spectroscopy (FT-IR) analysis, no significant chemical reactions were observed between asphalt and biodiesel derivatives. Previous studies have shown that biodiesel and its derivatives can increase the proportion of saturated and aromatic compounds in asphalt, which can prevent the accumulation of highly oxidized components in aged asphalt [17,18]. Recently, Pais et al. [19] investigated biodiesel and its derivatives produced using waste cooking oil (WCO) and animal fats, finding that the performance of asphalt mixtures containing biodiesel derivatives (dynamic modulus, phase angle, fatigue resistance, and rutting resistance) is compatible with that of mixtures containing conventional bio-asphalt [20].

In summary, regarding the characteristic that biodiesel and diesel can effectively reduce asphalt viscosity through mutual solubility, this paper employs viscosity experiments and the evaluation of mixture performance to comprehensively assess the effectiveness of biodiesel as a diluent. This provides a theoretical basis for further exploring the dilution effects of biodiesel and improving the optimized design of asphalt mixtures.

## 2. Materials and Methods

### 2.1. Raw Materials

#### 2.1.1. Biodiesel

Biodiesel from a biodiesel producer in Yunnan Province, China, is characterized by its golden-yellow color, clarity, and absence of visible impurities, with excellent fluidity. Its production process primarily involves using waste animal and vegetable oils (including gutter oil and chemically treated oils), which undergo impurity filtration and de-watering, followed by esterification and distillation to purify and obtain biodiesel. Biodiesel primarily contains lipids, alcohols, and acids, with some components chemically similar to the aromatic compounds and alkanes in asphalt, suggesting a good solubility of biodiesel in asphalt. The low viscosity and high lubricity of biodiesel effectively dilutes asphalt, while its low acid value reduces water damage to asphalt [20,21]. The main technical specifications of biodiesel are shown in Table 1.

#### 2.1.2. Diesel and Base Asphalt

For the cold patch asphalt formulation, in this paper, 70# road petroleum asphalt was used as the matrix asphalt for the preparation of cold patch asphalt from the Kunchu Expressway Highway Project (Yunnan Province, China), and commonly used #0 diesel (Yunnan Province, China) in road construction was selected as a diluent, blended with biodiesel to dilute the base asphalt to meet construction and workability requirements. The technical specifications of #0 (Yunnan Province, China) diesel and #70 base asphalt are shown in Table 2 and Table 3.

#### 2.1.3. Ancillary Additives

This study synthesized a cold patch asphalt additive (add) using epoxy resin and a curing agent as auxiliary additives. The main function was to reduce the viscosity of asphalt while enhancing the bonding effect between the diluted asphalt and aggregates. The selected epoxy resin was a Bisphenol A type from Fenghuang Brand (Jinan, China), and the curing agent used was a T31-modified amine curing agent (Zhengzhou, China). The main proportions are shown in Table 4 below.

## 3. Preparation of Cold Patch Asphalt

### 3.1. Pretreatment of Biodiesel

In biodiesel, fatty acid methyl esters (FAMEs) represent the largest component. This study processed biodiesel to produce biodiesel containing epoxy fatty acid methyl esters, enhancing the formation of epoxy systems and improving the performance of cold patch asphalt. The synthesis of epoxy fatty acid methyl esters occurs under acidic conditions, where formic acid (Jinan, China) reacts with hydrogen peroxide (Jinan, China) to form the oxidizing agent, peracetic acid. This subsequently reacts with the double bonds in fatty acid methyl esters as per the following main reaction equation [26]:

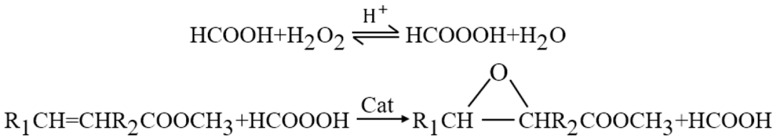


Process conditions were set as biodiesel/formic acid/hydrogen peroxide = 1:0.1:0.6. Biodiesel and formic acid were added to a three-necked flask and reacted with a magnetic stirrer to ensure thorough mixing. The reaction temperature was maintained at 60 °C, with hydrogen peroxide slowly added at the set temperature. After the addition was complete, the reaction was sustained for 4 h. Upon completion, the reaction mixture was transferred to a separatory funnel and left to stand until phase separation occurred. The aqueous layer was then discarded, and the pH was adjusted to approximately 7 with an alkali solution. Finally, the crude product was distilled under reduced pressure to obtain the desired product, which was biodiesel containing epoxy fatty acid methyl esters.

Epoxy fatty acid methyl esters are chemicals with excellent dispersibility, lubricity, and compatibility. They are effective dispersants widely used in coatings and also serve as biodegradable lubricants. Due to the high polarity and ring strain of the epoxy functional groups in their molecules, they exhibit exceptional reactivity [27]. This enables their critical role in various synthetic reactions, such as ring-opening and polymerization reactions, and their use in synthesizing materials like polyurethanes, polyesters, and epoxy resins [28].

### 3.2. Brookfield Viscosity Test

The viscosity and workability of cold patch asphalt depend on the proportion of the diluent used, and the judicious selection of diluent viscosity ensures excellent construction ease and long-term storability of the cold patch asphalt. In this study, four different mass ratios of diesel to biodiesel (90:10, 80:20, 70:30, 60:40) were used to prepare diluted asphalt. According to ASTM D 4402 “American Society for Testing and Materials. Standard Test Method for Viscosity Measurement of Bituminous Mixtures” [29], the viscosity of CPA at 20, 40, 60, and 80 °C was tested using an NDJ-1C Brookfield Rotational Viscometer (Changji, Shanghai, China) to preliminarily evaluate its workability. Additionally, the Brookfield viscosity of diesel, biodiesel, and auxiliary additives (add) at different temperatures was also tested, with results shown in Table 5.

From Table 4, it is evident that the viscosities of the various agents decrease as the temperature increases, and whether it is diesel, biodiesel, or auxiliary additives, their viscosities are significantly lower than that of the base asphalt. Diesel has traditionally been used as an asphalt diluent, while biodiesel provides excellent lubrication. Based on the principle of similar compatibility, their mutual solubility effectively reduces the viscosity of the base asphalt. Therefore, replacing diesel with biodiesel as a diluent for cold patch asphalt is a feasible method.

### 3.3. Preparation and Primary Selection of Cold Patch Asphalt

In this paper, #70 asphalt was selected as the matrix asphalt, #0 diesel and biodiesel were used as diluents, and auxiliary additives were used as thickeners to prepare CPA. The CPA was prepared using the following process: the matrix asphalt was first heated and insulated in an oven at 135 °C until it softened and flowed. The room temperature diluent was then added to the matrix asphalt in a set ratio, and the mixture was maintained at a temperature of 60–70 °C and stirred at 1000 rpm for 10 min using a high-speed stirrer to obtain diluted asphalt. After the diluted asphalt cooled to room temperature, the auxiliary additives, measured in a predetermined ratio, were added, and the mixture was sheared at 2000 rpm for 10 min using a high-speed shearing machine to finally obtain the prepared CPA.

Viscosity is commonly used as an indicator to determine whether CPA meets the requirements for construction and workability. In this paper, several different CPA formulations were designed based on different ratios of diesel to biodiesel and diluent proportions. Initially, these formulations were preselected through viscosity tests. To ensure a uniform application of the CPA on the aggregate, it is generally required that its rotational viscosity at 60 °C be between 0.8 and 1.6 Pa·s [30,31,32]. The Brookfield viscosity of the diesel and biodiesel was tested according to different mass ratios and different diluent-to-matrix asphalt ratios. To ensure the accuracy of the viscosity, each formulation was tested twice for viscosity. The specific formulation and test results of the CPA samples are shown in Figure 1. Here, biodiesel and diesel are represented by B and D, respectively, and the auxiliary additive is labeled as “add”. The Brookfield viscosities of CPA samples at 60 °C with diluent proportions of 20%, 22%, 24%, and 26% were tested.

According to Figure 1, the dilution effect of the biodiesel–diesel blend is better than that of pure diesel. When the amount of diluent increased from 20% to 26%, the Brookfield viscosity of the CPA decreased by about 61%. Compared to the auxiliary additives, the increase in biodiesel content has a minor impact on CPA’s viscosity. At the same time, the lowest viscosity is achieved when the ratio of biodiesel to diesel is 3:7 at diluent contents of 20%, 22%, and 24%. The Brookfield viscosity of the CPA samples at 60 °C should be between 0.8 and 1.6 Pa·s to meet the requirements for construction and workability, which corresponds to diluent contents of 20%, 22%, and 24%. Therefore, a 22% diluent content is primarily used for subsequent CPA testing and mixture preparation, with a viscosity range of 0.995–1.49 Pa·s, meeting the required CPA viscosity specifications.

### 3.4. FTIR Test Analysis

To understand the mechanisms of CPA action, this study utilized the Nicolet iS20 Fourier transform infrared spectrometer to analyze the infrared spectra of CPAL, determining the changes in the functional groups of the asphalt binder before and after diluent addition. First, the asphalt was dissolved in carbon disulfide ($CS_{2}$) at a concentration of 0.05 g/mL; then, the homogeneous solution was dropped onto a potassium bromide disk. Finally, the sample was dried under a mercury lamp and placed into the instrument for scanning from 4000 ${cm}^{-1}$ to 400 ${cm}^{-1}$, with 64 scans conducted. The IR spectra of which asphalt and CPAL exhibit are shown in Figure 2 below.

Infrared spectra can identify functional groups in asphalt by observing changes in characteristic peaks. Based on the comparison of the infrared spectra in Figure 2, the following conclusions can be drawn:

Significant characteristic peaks for matrix asphalt and CPA are observed near the wavenumbers of 2918, 2850, 1456, 1373, and 724 ${cm}^{-1}$. The absorption peak for hydroxyl (O-H) appears at 2918 ${cm}^{-1}$, and the asymmetrical vibration peak for methylene ($CH_{2}$) is seen at 2850 ${cm}^{-1}$. Symmetrical and asymmetrical stretching and bending vibrations of the aromatic ring (C=C) occur at 1456 and 1373 ${cm}^{-1}$, respectively, with the out-of-plane bending vibration absorption peak of trans-olefins (CH) appearing at 724 ${cm}^{-1}$ [33].

The infrared spectrum of CPA showed a distinct characteristic peak near the wave number of 1742 ${cm}^{-1}$, which exhibited a telescopic vibrational absorption peak of an ester group (C=O), whereas there was no carboxylate group absorption peak at 1705 ${cm}^{-1}$, which indicated that the epoxidized fatty acid methyl ester reacted with the carboxylate group in the CPA. In addition, the absorption peak of carboxylate appeared at 1167 ${cm}^{-1}$, which further indicates that the epoxidized fatty acid methyl ester in biodiesel reacted with the CPA [34,35].

A comparison of the infrared spectra of the matrix bitumen and CPA reveals that the fatty acids in biodiesel dissolve in the bitumen, resulting in depletion of the (C=O) group of the bitumen and the production of new ketone hydroxy acids. In addition, although the characteristic peaks of the matrix asphalt and CPA are basically the same and the wave positions are roughly adjacent to each other, new substances are also produced. Thus, preparation of CPA belongs to a complex physical reaction process [36].

## 4. Preparation and Testing of Cold Patch Asphalt Mixtures

The gradation of aggregates significantly affects the initial strength of CPAMs. Therefore, using a graded crushed stone skeleton can provide sufficient shear strength for CPAMs, while the continuous grading of the aggregate structure enhances the durability, workability, and rut resistance of the pavement, fulfilling strength requirements without susceptibility to damage [37,38]. In this paper, aggregates of 10–20 mm and 5–10 mm basalt (Kunming, China), as well as 0–5 mm limestone fillers (Kunming, China), were used to prepare the CPAM. The physical properties of these aggregates are presented in Table 6. A specific gradation design was developed, as shown in Figure 3, which represents the coarse and fine aggregate grading curves for the CPAM, meeting the requirements of the “Test Procedures for Asphalt and Asphalt Mixtures in Highway Engineering” (JTG E20-2011) [25].

The adhesion between the CPA and aggregates determines the performance of the CPAM and significantly affects the ease of application and the durability of potholes. Poor adhesion can result in insufficient bonding between the mixtures, leading to a loose CPAM after compaction, poor stability, and ultimately, the failure of pothole repairs [30]. Therefore, the adhesion between the CPAM and aggregates is a core indicator of the CPAM’s adequacy, making it essential to study this property.

### 4.1. Bonding Test

In the adhesion tests discussed in this paper, it is necessary to consider not only the adhesion properties of CPA but also to determine its optimal dosage. If the CPA content is insufficient, it can lead to an uneven coating of the CPAM, resulting in uncovered white aggregates and inadequate mixture bonding, making it difficult to achieve high strength. If the CPA content is too high, it can lead to excessive oiliness in the CPAM and the exudation of CPA, resulting in issues such as agglomeration [31,39].

In this study, based on the grading design discussed above, five different CPA dosages of 4.7%, 5.2%, 5.7%, 6.2%, and 6.7% were selected through adhesion tests to determine the optimal dosage for ensuring the best performance of the CPAM. The aggregate grading shown in Figure 4 was mixed using the same process, with each CPAM sample batch prepared with 500 g of sample. Then, 50 g was taken and placed in boiling water, maintained at boiling with uniform stirring for 2 min. Next, 100 pieces of aggregate were randomly selected and placed on white paper to check whether the aggregates were completely covered by CPA. The number of completely covered aggregates, denoted as N, and their adhesion rate were calculated as shown in Equation (1).
(1)$$P_{b}=\frac{N}{100}\times 100\%$$

A higher adhesion rate ($P_{b}$) indicates better CPA adhesion, with the results shown in Table 7.

Table 7 reveals that when the CPA content is 4.7% and 5.2%, the coating rate of the CPAM is significantly lower with a large presence of white material, indicating that a lower CPA content results in a thinner asphalt film on the aggregate surface. When the CPA content is 5.7% and 6.2%, the aggregate surface becomes oily and glossy, with a more uniform asphalt film and a moderately wet state. At a CPA content of 6.7%, the aggregate surface has an excess of CPA, resulting in asphalt flow, as shown in Figure 4 for different CPA contents. Additionally (Red circles indicate aggregates in the aggregate that are not fully coated with asphalt), the auxiliary additive effectively enhances the adhesiveness of CPA. CPAMs without the additive completely detach after boiling in water, whereas those with the additive show significantly improved adhesion. Therefore, the optimal CPA content is determined to be 6.2%.

### 4.2. Strength Growth Pattern of Epoxy Cold Patch Asphalt Mixtures

Considering the mechanical performance requirements of CPAMs, along with other performance indicators, this study employs a matrix asphalt system comprising an epoxy system = 80:20, diluent content of 22% (biodiesel/diesel = 30:70), and an asphalt usage of 6.2% to explore the strength growth patterns and curing trends in CPAMs.

As shown in Figure 5, the strength of the CPAM increases rapidly during the first 10 days of curing at room temperature. After 10 days of curing, it reaches its maximum strength, with a strength of up to 11.87 kN, meeting the requirements for HMA Marshall stability (≥8 kN) specified in the “Technical Specification for Highway Asphalt Pavement Construction” (JTG F40-2004) [40]. The stability of the CPAM was fitted using the Box Lucas model in Origin, with the function form shown in Equation (2). The Box Lucas model, due to its simple exponential form, can theoretically be used for the curve fitting of any process that exhibits a single-phase change without an offset in exponential variation.
(2)$$y=a\;(1-e^{bx})$$

The final stability fit curve obtained is
$$y=12.31\;(1-e^{-0.3687x})$$
where *a* and *b* are the parameters to be fitted.

Its fit R^2^ = 0.9236. The coefficient of determination (R^2^) is an important indicator for measuring the goodness of fit of a model, which reflects how well the model fits the experimental data. A value of R^2^ closer to one indicates a better fit of the model to the data. The innovation of the Box Lucas model lies in its ability to accurately capture the nonlinear behavior of materials under dynamic loading through a simple exponential function form, accurately capturing the nonlinear behavior of materials under dynamic conditions. Therefore, the growth strength of the epoxy cold patch asphalt mixture in this paper conforms to the strength growth law, and the growth rate is stable.

### 4.3. Rutting Test

The high-temperature stability of asphalt mixtures refers to their ability to resist deformation flow at high temperatures, and dynamic stability is a crucial indicator of this high-temperature stability. This paper investigates the effect of biodiesel on the high-temperature stability of the CPAM through rutting tests. Rut specimens were prepared according to the method used for CPAMs. Considering the loose initial state of diluted asphalt-type CPAMs, this preparation ensures that the early performance of the test specimens is representative. The specimens were initially shaped using a rutting wheel press with four back-and-forth passes, then left in their molds in the natural environment for three days. Subsequently, the specimens were placed in an oven at 110 °C for continuous heating for 12 h, followed immediately by an additional eight passes with the rutting wheel press. After seven more days of curing in the natural environment, finally, the specimens were tested using an automatic rutting tester (Shanghai Changji SYD-719 Automatic Rutting Tester, Shanghai, China). The dynamic stability was measured under test conditions of 60 °C to evaluate its resistance to deformation in high-temperature environments [41]. The wheel load was set at 0.7 MPa, and the average of three repetitions was recorded to assess the dynamic stability of the CPAM (DS, cycles/mm). The tests were conducted according to the “Standard Test Methods of Bitumen and Bituminous Mixtures for Highway Engineering” (JTG E20-2011) [25]. The dynamic stability calculation formula is shown in Equation (3).
(3)$$DS=\frac{(t_{1}-t_{2})\times N}{d_{2}-d_{1}}\times C_{1}\times C_{2}$$
where DS (cycle/mm) represents the dynamic stability of the asphalt mixture; $d_{1}$ (mm) is the deformation at time $t_{1}$ (min); $d_{2}$ (mm) is the deformation at time $t_{2}$ (min); $C_{1}$ (equals 1.0 in this study) is the machine type coefficient; $C_{2}$ (equals 1.0 in this study) is the specimen coefficient; and N (equals 42 cycles/minute) is the speed of the wheel’s back-and-forth motion.

The experimental results of this study are shown in Figure 6. Using the revised testing methods and the effect of auxiliary additives, the dynamic stability of each group met the requirements of the hot mix asphalt rutting test, with values ≥ 800 (times/mm). As the incorporation of biodiesel increased, the dynamic stability of the CPAM showed an upward trend compared to when only diesel was used, indicating that the addition of biodiesel progressively enhanced the high-temperature deformation resistance of the asphalt mixture. At a biodiesel content of 40%, the highest dynamic stability was achieved at 9027 (times/mm), a 94.7% improvement in high-temperature performance over the use of diesel alone as a diluent, and about seven times the performance of hot mix asphalt. The epoxy groups in the CPAM react with the curing agent to form a three-dimensional cross-linked structure [42], which embeds the asphalt in a network structure, significantly enhancing the mixture’s resistance to permanent deformation.

### 4.4. Submerged Marshall Test

Asphalt pavement water damage refers to the gradual penetration of water from the surface of the asphalt film into the aggregate under the repeated action of vehicle loads, leading to the detachment of the asphalt film and separation from the aggregate, causing damage to the pavement. When biodiesel is added to asphalt, the epoxy fatty acid methyl esters in the biodiesel, acting together with the epoxy resin, significantly increase the adhesion between the asphalt and the aggregate, enhancing the water stability of the CPAM [43,44]. The soaked Marshall test is one of the important tests for evaluating the water stability of diluted asphalt mixtures. The test was conducted according to the “Highway Engineering Asphalt and Asphalt Mixture Test Specifications” (JTG E20-2011) [25].

As shown in Figure 7, when the biodiesel content increases from 10% to 40%, the corresponding 0 h Marshall stability increases from 5.59 kN to 8.1 kN, and the 48 h Marshall stability increases from 4.8 kN to 7.39 kN, while the residual stability rises from 85.9% to 91.3%. This indicates that as the epoxy groups continue to increase, the epoxy system gradually forms, and the water damage resistance of the CPAM gradually improves. The increase in biodiesel content also increased the cross-linking points in the asphalt, continually improving the CPAM’s water stability.

## 5. Conclusions

This paper assesses the feasibility of using pretreated biodiesel to prepare CPA and CPAMs. The mechanical properties of CPA and the changes in microstructure during the curing process were studied through rotational viscosity tests, adhesion tests, strength growth tests of cold patch asphalt mixtures (CPAMs), rutting tests, and water immersion Marshall tests that were conducted to analyze the impact of biodiesel on the adhesion and performance of a CPAM. The results indicate that it is feasible to use partially pretreated biodiesel as a substitute for diesel as a diluent. The main conclusions are as follows:(1)The biodiesel used in this study enhanced the coverage and workability due to its excellent lubrication performance, thereby reducing the addition of diesel and increasing the epoxy group content in the CPAM. The formation of a three-dimensional cross-linked structure in the epoxy system facilitated the bonding between asphalt and the aggregate, enhancing the adhesive properties of the diluted asphalt mixture while ensuring its strength for road surface applications.(2)With the incorporation of biodiesel, the dynamic stability of the CPAM shows an increasing trend compared to using only diesel. This indicates that the addition of biodiesel gradually enhances the ability of the asphalt mixture to resist high-temperature deformation. The maximum dynamic stability is achieved when the biodiesel content reaches 40%, reaching 9027 (times/mm), which represents an improvement of approximately 94.7% in high-temperature performance compared to 4636 (times/mm) when using only diesel as a diluent and an approximately seven times better high-temperature performance compared to hot mix asphalt mixtures.(3)When the biodiesel content increases from 10% to 40%, the corresponding 0 h Marshall stability increases from 5.59 kN to 8.1 kN and the 48 h Marshall stability increases from 4.8 kN to 7.39 kN, while the residual stability rises from 85.9% to 91.3%. This indicates that as the epoxy groups continue to increase, the epoxy system gradually forms, and the water damage resistance of the CPAM gradually improves. With the increase in biodiesel content, there are more cross-linking points in the asphalt, leading to improved water stability of the CPAM.(4)The biodiesel used in this paper can reduce the solvent (diesel) consumption of CPAMs, significantly improving their road performance. The performance of CPAM samples with biodiesel/diesel ratios of 30:70 and 40:60 was superior to those with a 0:100 ratio, demonstrating that substituting diesel with pretreated biodiesel, which contains epoxy fatty acid methyl esters, can better promote the formation of the epoxy system, not only reducing the dilution of asphalt but also significantly enhancing the road performance of the CPAM.(5)In summary, the epoxy cold patch asphalt mixture using processed biodiesel as a diluent demonstrates outstanding performance and broad application prospects. Processed biodiesel not only effectively reduces the viscosity of asphalt and improves its adhesion to aggregates, but also outperforms traditional asphalt mixtures in durability, water damage resistance, and performance. The use of this innovative material not only enhances construction convenience but also meets the demands of sustainable development, reflecting environmental protection principles. In the future, the epoxy cold patch asphalt mixture using processed biodiesel as a diluent will play a greater role in road construction and maintenance, promoting the development of green building materials.

## Figures and Tables

**Figure 1 materials-17-05566-f001:**
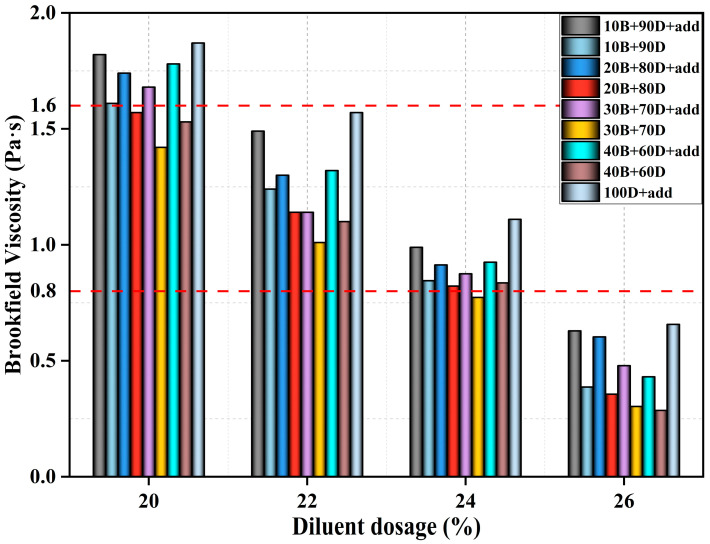
Brookfield viscosity test results of CPA at 60 °C.

**Figure 2 materials-17-05566-f002:**
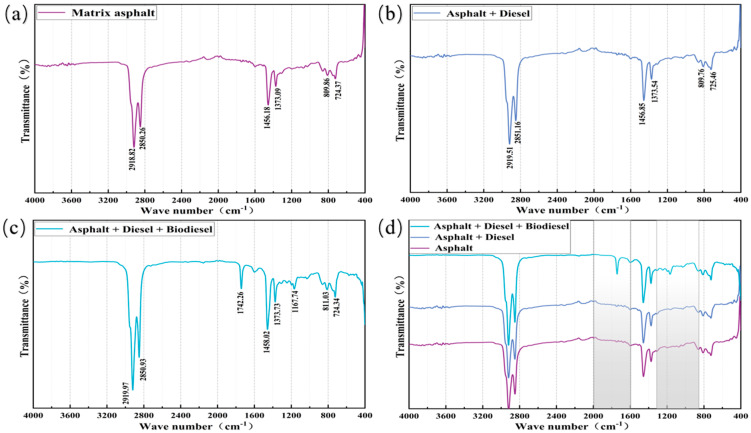
Infrared spectra of matrix asphalts and CPA. (**a**) Matrix asphalt. (**b**) Matrix asphalt + diesel. (**c**) Matrix asphalt + diesel + biodiesel. (**d**) Comparison of the results of the IR tests of the three types of asphalt.

**Figure 3 materials-17-05566-f003:**
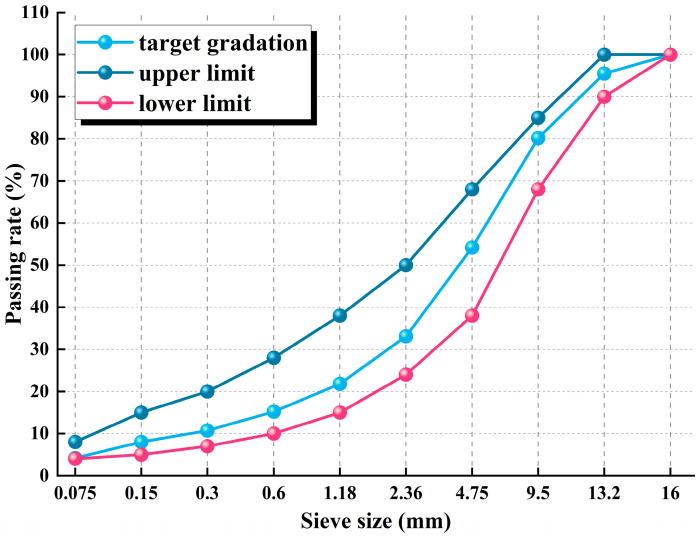
Aggregate grading curves of prepared mixes.

**Figure 4 materials-17-05566-f004:**
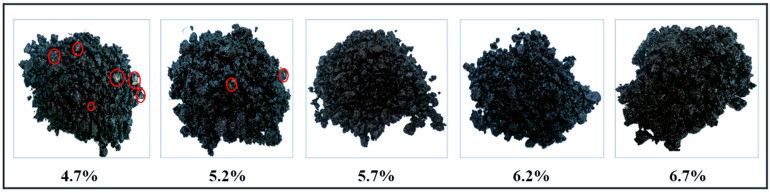
Appearance of CPAM with different CPA contents.

**Figure 5 materials-17-05566-f005:**
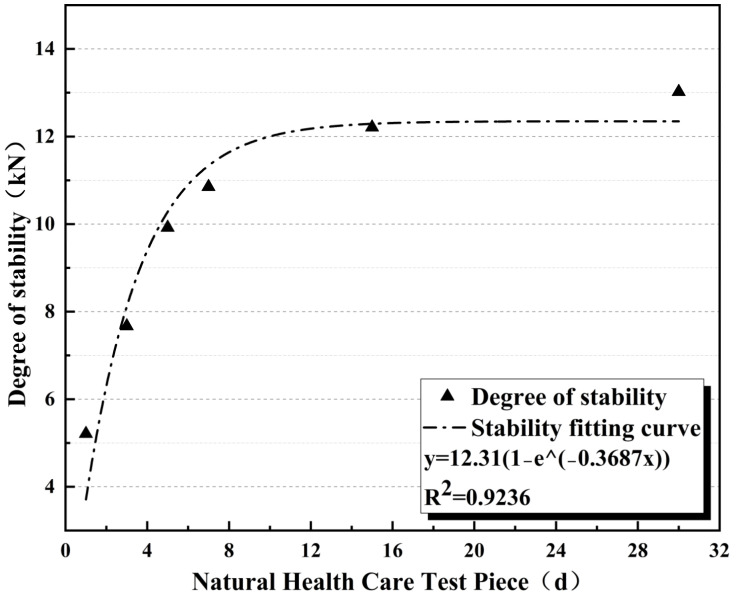
Stability growth fitting curve.

**Figure 6 materials-17-05566-f006:**
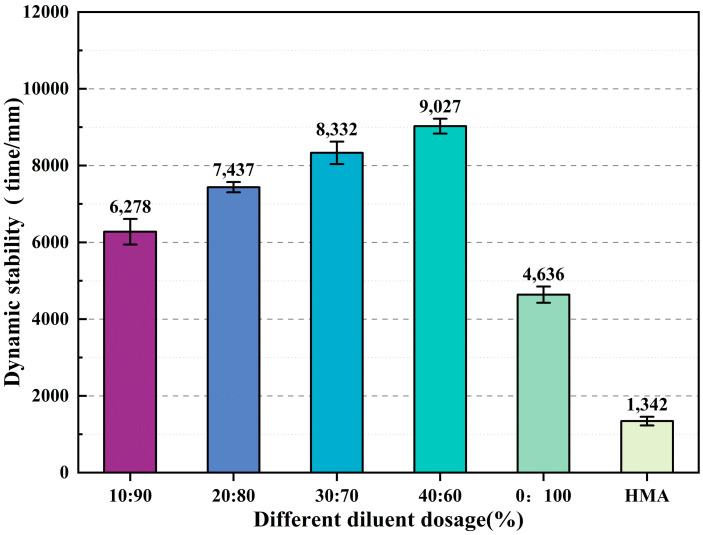
High-temperature performance test results with different diluent dosages.

**Figure 7 materials-17-05566-f007:**
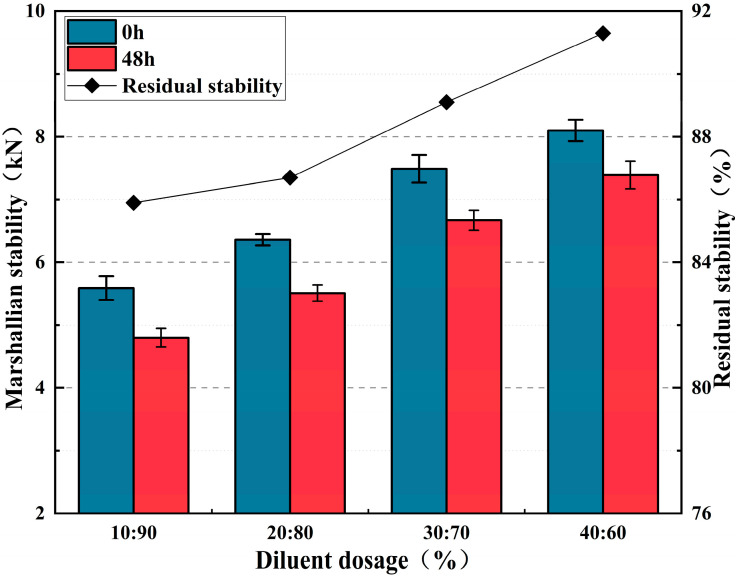
Marshall test results with different diluent dosages.

**Table 1 materials-17-05566-t001:** Technical specifications of biodiesel.

	Test Results
$20\;{}^{\circ}C\;\mathrm{specific}\;\mathrm{gravity}/kg/m^{3}$	0.881
flash point/°C	135
acid value/mgKOH/100 mL	1.03
cetane number/CN	58
sulfur content/%	≤0.01
iodine value/g/100 g	106

**Table 2 materials-17-05566-t002:** Technical specifications of 0# diesel oil.

Test Items	Test Results	Chinese Standard
$\mathrm{standard}\;\mathrm{density}\;(20\;{}^{\circ}C)/kg/m^{3}$	0.839	GB/T 1884 [22]
acid value/mg KOH/100 mL	4.75	GB/T 258 [23]
kinematic viscosity (20 °C)/mm^2^/s	4.35	GB/T 265 [24]

**Table 3 materials-17-05566-t003:** Technical specifications of 70# matrix asphalt.

Test Items	Test Results	Chinese Standard (JTG E20-2011) [25]
penetration (25 °C)/0.1 mm	63.8	T0604
ductility (10 °C)/cm	46	T0605
softening point/°C	49.4	T0606
dynamic viscosity (60 °C)/Pa·s	3.11	T0620
flash point/°C	>300	T0611
aging process (163 °C, 5 h)		
mass loss/%	0.18	T0609
penetration (25 °C)/0.1 mm	55.4	T0604
ductility (10 °C)/cm	13	T0605

**Table 4 materials-17-05566-t004:** Main ratios used in this paper.

No.	Biodiesel (%)	Diesel (%)	Epoxy Resin (%)	Curing Agent (%)
1	10	90	40	40
2	20	80	40	40
3	30	70	40	40
4	40	60	40	40
5	0	100	40	40

**Table 5 materials-17-05566-t005:** Thinner Brookfield viscosity test results.

Materials	20 °C	40 °C	60 °C	80 °C
diesel (mPa·s)	2.5	1.5	0.8	0.3
biodiesel (mPa·s)	41.5	19.0	6.0	2.5
add (mPa·s)	107.0	45.5	21.0	6.5

**Table 6 materials-17-05566-t006:** Physical properties of aggregates.

Grain Size (mm)	10–20	5–10	3–5	0–3
apparent density (g/cm^3^)	2.878	2.865	2.843	2.826
water absorption (%)	0.52	0.52	0.38	0.38
crushing value (%)	11.22	11.22	-	-

**Table 7 materials-17-05566-t007:** Adhesion rate of CPA.

m_B_:m_D_:m_add_	Coating Rate (%)				
	CPA Dosage (%)				
	4.7	5.2	5.7	6.2	6.7
10:90:30	93	97	99	100	100
20:80:30	93	96	100	100	100
30:70:30	91	96	99	100	100
40:60:30	92	98	99	100	100
40:60:0	0	0	0	0	0

## Data Availability

Raw/processed data required to reproduce these findings cannot be shared at this time as the data also forms part of an ongoing study.

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
