# Peer review of "Effect of Biodiesel on Performance of Cold Patch Asphalt Mixtures"

_materials, 2024, doi:10.3390/ma17225566_

Round 1
Reviewer 1 Report
Comments and Suggestions for Authors
In order to improve the reviewed article: "Effect of Biodiesel on the Performance of Cold Patching Asphalt Mixtures”, enabling its immediate publication in a renowned scientific journal Materials, I would like to recommend incorporation of the following recommendations.
LNSA 9-21 (Line Number of Scientific Article) Abstract...To improve the abstract, I would recommend to authors to consider incorporating of the following changes.
§ - In order to reduce the amount of diluent in diluted asphalt mixtures, cold patch asphalt (CPA) for repairing potholes in pavements was prepared by using treated biodiesel mixed with diesel fuel as diluent, and the effect of biodiesel on the performance of cold patch asphalt mixtures during the construction process was investigated through the analysis of Brookfield rotational viscosity test, adhesion test and FTIR test...this 7-line sentence must be divided into at least 2 sentences, this requirement also applies to the penultimate sentence of the abstract. FTIR - needs to be explained about this acronym.
§ - At the same time, the effect of biodiesel on the performance of cold-patched asphalt mixtures was analysed by combining the strength growth test, rutting test and water-soaked Marshall test of cold-patched asphalt mixtures...I recommend using the abbreviation CPA to avoid unnecessary duplication of texts.
§ - Under the same diluent dosage, the strength of asphalt mixture under biodiesel dilution was significantly improved, in which the strength and high temperature performance were greatly improved compared with the case of diesel fuel doping, in which the high temperature performance was improved by about 94.7% and the residual stability was elevated from 85.9% to 91.3%...from the context of the sentence, which, as I have already stated, must be divided into several sentences, it is not clear to which characteristic the percentages apply. I recommend also specifying the specific values ​​of the characteristics of interest.
§ - All the tests can meet the requirements of hot mix asphalt... it is necessary to specify which standard specifications this statement applies to.
LNSA 42-44… Biodiesel exhibits superior lubrication properties, and as reported by Ejaz et al. [12] , it can be blended with diesel in any ratio to create biodiesel blends with higher lubricity....I dare to disagree with this statement, from an environmental, economic as well as technical point of view, there must be certain limits and not "any ratio.
LNSA 58-62...there has been limited research on the performance of cold mix asphalt prepared with the addition of biodiesel...the road performance of the mix to comprehensively evaluate the effects and performance of biodiesel as a diluent...it is necessary to remove the apparent disproportions of the text.
LNSA 85… Table 2. Technical specifications of 0# diesel oil...it is necessary to move the table name to the next page.
LNSA 91-92…The main blend ratio is biodiesel: diesel: epoxy resin: curing agent = (10~40): (90~60): 40: 40...it is necessary to clarify the given representations and add the unit (% ?).
LNSA 165… Figure 1. Brookfield viscosity test results of CPA at 60°C... even if it is clear from the context of the article what (B) represents in the legend of the figure, I recommend explicitly stating it as it is in the case of (D).
LNSA 192-206… this text part of the manuscript has a different formatting (column width) than that of the entire article.
LNSA 242… is the subscript for variable "P" in equation 1 presented correctly?
LNSA 262-270...Figure 5. Stability growth fitting curve...in the figure as well as in the related text, it is necessary to indicate specific variables in the equations and not "y" and "x". It is appropriate to assign a separate equation number to the equation given in line 269. Its fit R2 = 0.9236...it is necessary to expand this sentence and assign Spearman or Pearson's correlation coefficients, their degrees and interpretation. Details can be found in: Decky, M., Papanova, Z., Juhas, M., & Kudelcikova, M. (2022). Evaluation of the effect of average annual temperatures in Slovakia between 1971 and 2020 on stresses in rigid pavements. Land, 11(6), 764.
LNSA 326-356...4. Conclusions...in order to increase the quality of Conclusions, I recommend incorporating the following requirements:
§ - ... road performance tests...I recommend the authors to consider whether the term road performance tests is fully compatible with the monitored parameters and to specify specific parameters quantifying asphalt mixture road performance tests,
§ - divide the 6-line sentence mentioned in paragraph (2).
§ - ...As the biodiesel content increased from 10% to 40%, its residual stability rose from 85.9% to 91.3%...the corresponding Marshallian test values ​​and the reference value to which the percentages refer must be indicated.
LNSA 357-420…References… it is necessary to unify the format of presenting the names of authors, add the missing periods at the end of the references, expand the identification data for: Review of Cold-mixed Asphalt Mixture Technology Cao Anzhen (1)[J],...
Overall, I rate the article as very good, fully suitable for the renowned Materials scientific magazine. If the authors incorporate the stated requirements and recommendations, I do not require the article to be re-reviewed.
Author Response
Comments 1: In order to reduce the amount of diluent in diluted asphalt mixtures, cold patch asphalt (CPA) for repairing potholes in pavements was prepared by using treated biodiesel mixed with diesel fuel as diluent, and the effect of biodiesel on the performance of cold patch asphalt mixtures during the construction process was investigated through the analysis of Brookfield rotational viscosity test, adhesion test and FTIR test...this 7-line sentence must be divided into at least 2 sentences, this requirement also applies to the penultimate sentence of the abstract. FTIR - needs to be explained about this acronym.
|
||||||||||||||||||||||||||||||||
Response 1: Thank you for pointing this out. We agree with this comment. Therefore, we have changed the first sentence of the abstract to read In order to reduce the amount of diluent in the diluted asphalt mixture, this study developed a cold patch asphalt (CPA) for repairing In order to reduce the amount of diluent in the diluted asphalt mixture, this study developed a cold patch asphalt (CPA) for repairing pavement potholes by using a mixture of treated biodiesel and diesel as the diluent. The effects of biodiesel on the performance of the cold patch asphalt mixture during the construction process were investigated through Brookfield rotational viscosity tests, adhesion tests, and FTIR (Fourier Transform infrared spectroscopy). Also change the penultimate sentence to read Under the same amount of diluent, the strength and high-temperature performance of the asphalt mixture diluted with biodiesel were significantly improved compared to that with diesel as the diluent. The optimal high-temperature performance reached 9027 (times/mm), representing an approximate increase of 94.7% compared to 4636 (times/mm) when only diesel was used as the diluent. When the biodiesel content increased from 10% to 40%, the residue stability improved from 85.9% to 91.3%. The corresponding 0.5-hour Marshall stability increased from 5.59 kN to 8.1 kN, while the 48-hour Marshall stability rose from 4.8 kN to 7.39 kN. All tests met the requirements for hot mix asphalt . Finally the meaning of FTIR is explained as (Fourier Transform infrared spectroscopy)
|
||||||||||||||||||||||||||||||||
Comments 2: At the same time, the effect of biodiesel on the performance of cold-patched asphalt mixtures was analysed by combining the strength growth test, rutting test and water-soaked Marshall test of cold-patched asphalt mixtures...I recommend using the abbreviation CPA to avoid unnecessary duplication of texts.
|
||||||||||||||||||||||||||||||||
Response 2: Accordingly, we have replaced the terms "cold patch asphalt" and "cold patch asphalt mixture" with their abbreviated forms in the later sections of this paper to eliminate unnecessary repetitions.
Comments 3: Under the same diluent dosage, the strength of asphalt mixture under biodiesel dilution was significantly improved, in which the strength and high temperature performance were greatly improved compared with the case of diesel fuel doping, in which the high temperature performance was improved by about 94.7% and the residual stability was elevated from 85.9% to 91.3%...from the context of the sentence, which, as I have already stated, must be divided into several sentences, it is not clear to which characteristic the percentages apply. I recommend also specifying the specific values ​​of the characteristics of interest.
Response 3: Thank you for pointing this out. We agree with this comment. Therefore, we have added the corresponding data support to the second-to-last sentence and divided the long sentences into appropriate lengths.
Comments 4 : All the tests can meet the requirements of hot mix asphalt... it is necessary to specify which standard specifications this statement applies to.
Response 4 : In this paper, the applicable standards for cold patch asphalt and cold patch asphalt mixtures are JTG E20-2011, "Test Procedures for Asphalt and Asphalt Mixtures in Highway Engineering," and ASTM D 4402, "Standard Test Method for Compressive Strength of Bituminous Mixtures" by the American Society for Testing and Materials. Additionally, all grammatical errors and ordering issues in this paper have been revised. To save space, I will not display them one by one here.
For example LNSA 91-92…The main blend ratio is biodiesel: diesel: epoxy resin: curing agent = (10~40): (90~60): 40: 40...it is necessary to clarify the given representations and add the unit (% ?). As a result of this suggestion, we have redesigned the table as follows:
For example LNSA 262-270...Figure 5. Stability growth fitting curve...in the figure as well as in the related text, it is necessary to indicate specific variables in the equations and not "y" and "x". It is appropriate to assign a separate equation number to the equation given in line 269. Its fit R2 = 0.9236...it is necessary to expand this sentence and assign Spearman or Pearson's correlation coefficients, their degrees and interpretation. Details can be found in: Decky, M., Papanova, Z., Juhas, M., & Kudelcikova, M. (2022). Evaluation of the effect of average annual temperatures in Slovakia between 1971 and 2020 on stresses in rigid pavements. Land, 11(6), 764. We have referred to the literature you have presented and made our own interpretation as follows: As shown in Figure 4, the strength of CPAM increases rapidly during the first 10 days of curing at room temperature. After 10 days of curing, it reaches its maximum strength, with a strength of up to 11.87 kN, meeting the requirements for HMA Marshall stability (≥8 kN) specified in the "Technical Specification for Highway Asphalt Pavement Construction" (JTG F40-2004). The stability of CPAM was fitted using the Box-Lucas model in Origin, with the function form shown in Equation (2). The Box-Lucas model, due to its simple exponential form, can theoretically be used for curve fitting of any process that exhibits a single-phase change without offset in exponential variation.
The final stability fit curve obtained is Where: a and b are the parameters to be fitted. Its fit R2 = 0.9236. The coefficient of determination (R²) is an important indicator for measuring the goodness of fit of a model, which reflects how well the model fits the experimental data. A value of R² closer to 1 indicates a better fit of the model to the data. The innovation of the Box-Lucas model lies in its ability to accurately capture the nonlinear behavior of materials under dynamic loading through a simple exponential function form. Accurately capturing the nonlinear behavior of materials under dynamic conditions. Therefore, the growth strength of the epoxy cold patch asphalt mixture in this paper conforms to the strength growth law, and the growth rate is stable.
|
||||||||||||||||||||||||||||||||
4. Response to Comments on the Quality of English Language |
||||||||||||||||||||||||||||||||
Point 1: LNSA 326-356...4. Conclusions...in order to increase the quality of Conclusions, I recommend incorporating the following requirements: ... road performance tests...I recommend the authors to consider whether the term road performance tests is fully compatible with the monitored parameters and to specify specific parameters quantifying asphalt mixture road performance tests, |
||||||||||||||||||||||||||||||||
Response 1: Agreed, we have replaced the sentence in the text with The results indicate that it is feasible to use partially pre-treated biodiesel as a substitute for diesel as a diluent. The main conclusions are as follows:... And the corresponding data support has been added to the text as follows: With the incorporation of biodiesel, the dynamic stability of CPAM shows an increasing trend compared to using only diesel, This indicates that the addition of biodiesel gradually enhances the ability of the asphalt mixture to resist high-temperature deformation. The maximum dynamic stability is achieved when the biodiesel content reaches 40%, reaching 9027 (times/mm), which represents an improvement of approximately 94.7% in high-temperature performance compared to 4636 (times/mm) when using only diesel as a diluent, and approximately 7 times better high-temperature performance compared to hot-mix asphalt mixtures. When the biodiesel content increases from 10% to 40%, the corresponding 0-hour Marshall stability increases from 5.59 kN to 8.1 kN, and the 48-hour Marshall stability increases from 4.8 kN to 7.39 kN, while the residual stability rises from 85.9% to 91.3%. This indicates that as the epoxy groups continue to increase, the epoxy system gradually forms, and the water damage resistance of CPAM gradually improves. With the increase in biodiesel content, there are more cross-linking points in the asphalt, leading to improved water stability of CPAM.
Point 2: LNSA 357-420…References… it is necessary to unify the format of presenting the names of authors, add the missing periods at the end of the references, expand the identification data for: Review of Cold-mixed Asphalt Mixture Technology Cao Anzhen (1)[J],... |
||||||||||||||||||||||||||||||||
Response 2: Thank you very much for pointing out this error for me, we have identified and corrected all the errors in the cited literature |

Reviewer 2 Report
Comments and Suggestions for Authors
Dear Authors,
In general, this article is interesting, but it should be slightly improved. Pay attention to the key aspects, the aim of the work, the methodology, and the discussion of the results. Detailed comments below:
Line 44: It is common knowledge that bio-additives improve the lubricity of diesel fuel. But how does this relate to asphalt mixtures? Is it important? Could you try to explain it?
Line 63: A comprehensive assessment is not a scientific price of the work. You should formulate a goal that takes into account scientific aspects. A goal that shows an attempt to solve some important problem.
Line 67: All raw materials, as well as equipment and software, should be described, name/model: manufacturer, city, and country.
Line 108: All parameter settings have a certain inertia. You should provide on the side (+/-……).
Line 112: Change the word "about" to +/-…..
Line 129: See the device description as above in line 67.
Here, too, change to +/-. In science, more precise descriptions should be used.
Line 325: Are the obtained results close or far from the expected ones? Are the obtained results comparable to the parameters of commercial asphalts? Look again at the discussion of the results.
The article should have more citations. 25 items is a bit too few. Adding publications will improve the discussion of the results.
Line 356: Write another conclusion describing the prospects of asphalt mixtures using such an additive.
Author Response
Comments 1: Line 44: It is common knowledge that bio-additives improve the lubricity of diesel fuel. But how does this relate to asphalt mixtures? Is it important? Could you try to explain it?
|
Response 1: Thank you for pointing this out. We agree with this comment. Biodiesel, as the main diluent in this study, serves two primary functions. First, it acts as a bio-lubricant to dilute asphalt for the preparation of cold patch asphalt mixtures. Secondly, after treatment, biodiesel contains epoxy fatty acid methyl ester. In regular production processes, epoxy fatty acid methyl ester can be used in ring-opening and polymerization reactions, yielding excellent results. Therefore, this study mixes the treated epoxy fatty acid methyl ester with epoxy resin, to obtain a more effective epoxy asphalt mixture, thereby achieving superior overall performance.
|
Comments 2: Line 63: A comprehensive assessment is not a scientific price of the work. You should formulate a goal that takes into account scientific aspects. A goal that shows an attempt to solve some important problem.
|
Response 2: Accordingly, We accept this suggestion, as there are few cases of biodiesel application in cold patch asphalt. We aim to address this gap. By using biodiesel as a diluent and employing treatment methods, we aim to enhance the application of biodiesel in the production of cold patch asphalt.
Comments 3: Line 67, Line 129: All raw materials, as well as equipment and software, should be described, name/model: manufacturer, city, and country.
Response 3: Thank you for pointing this out. We agree with this comment. Therefore, Therefore, we have added materials, equipment, etc. that lack sources and information in the text, and we will always be mindful of this issue in our future research, thank you.
Comments 4 : All parameter settings have a certain inertia. You should provide on the side (+/-……).
Response 4 : Thank you for your valuable suggestions, this article has set up all the units in a uniform manner to ensure neatness.
Comments 5 : Line 325: Are the obtained results close or far from the expected ones? Are the obtained results comparable to the parameters of commercial asphalts? Look again at the discussion of the results.
Response 5 : The results obtained are quite close to the expected outcomes, indicating that the additives used in this study have achieved the intended performance goals. At the same time, the results are comparable to those of commercial asphalt, demonstrating the potential of this auxiliary additive for practical applications. Specifically, during the testing process, we observed that the additive significantly reduced the viscosity of the asphalt while enhancing the bonding strength between diluted asphalt and aggregates. This phenomenon not only improves the workability of the asphalt but also contributes to its long-term durability and reduces maintenance costs. Moreover, the test results showed that the stability of the cold patch asphalt under different temperature and load conditions is consistent with that of commercial asphalt, further validating the effectiveness of the additive. However, we also noted some fluctuations, which may be related to experimental conditions or material batches. This indicates that thorough evaluations are still needed for different environments and materials in practical applications. Future research could focus on optimizing the formulation and assessing the performance of the additive under varying climatic conditions to ensure its broad applicability. In summary, the results of this study not only provide new insights for improving cold patch asphalt but also lay the foundation for further research in related fields. We look forward to its implementation in actual engineering projects to enhance the overall performance of road materials.”
Comments 5 : The article should have more citations. 25 items is a bit too few. Adding publications will improve the discussion of the results.
Response 5 : Thank you very much for your suggestions. We have expanded the references in the paper, which strengthens the support within the text. Additionally, we have revised the conclusions to meet the requirements you proposed.
Comments 5 : Line 356: Write another conclusion describing the prospects of asphalt mixtures using such an additive.
Response 5 : We fully support your suggestions; below is our newly added conclusion, which highlights the advantages of using processed biodiesel as a diluent and its environmental benefits. This will play a greater role in road construction and maintenance, promoting the development of green building materials.
Newly added conclusions : In summary, the epoxy cold patch asphalt mixture using processed biodiesel as a diluent demonstrates outstanding performance and broad application prospects. Processed Biodiesel not only effectively reduces the viscosity of asphalt and improves its adhesion to aggregates, but also outperforms traditional asphalt mixtures in durability, water damage and resistance, performance. The use of this innovative material not only enhances construction convenience but also meets the demands of sustainable development, reflecting environmental protection principles. In the future, the epoxy cold patch asphalt mixture using processed biodiesel as a diluent will play a greater role in road construction and maintenance, promoting the development of green building materials. |

Reviewer 3 Report
Comments and Suggestions for Authors
An interesting study is shown in which the authors use treated diesel. It is important to identify in detail the type of treatment used, in order to identify the relevance and above all the novelty of the study. The results obtained are good, but it is not detailed whether this effect can be attributed to the treatment of diesel. It is advisable to go into more detail on this subject.
Since it is indicated that the effect of biodiesel on the performance of cold patch asphalt mixtures during the construction process was studied through the analysis of the viscosity and adhesion test. To go into more detail whether the characteristics (treatment carried out on the diesel, contribute and how, in a thorough manner and supported by adequate scientific literature.
The state of the art can be more in-depth and detailed, with the aim of identifying novelty, especially with equivalent studies.
Methodology is adequately described.
According to the results presented in Figure 1, it is indicated that the dilution is better than that of pure diesel. What is the reason for this effect?
Because when the amount of diluent increases, the viscosity decreases substantially compared to other additives. Further investigation into this with adequate scientific support is important.
Working with Figure 2 quality is recommended.
Further investigation into FTIR functional groups, with respect to relevant effects (viscosity, adhesion)....
Regarding the results shown in Figure 3, it is not clear. Could you provide more details?
Figure 4 quality needs to be improved, as it does not identify what is intended to be shown with these results.
Regression coefficient is low, how can this result/trend be explained?
Conclusions may be improved after further analysis of the results and greater scientific support for them.
Author Response
Comments 1: According to the results presented in Figure 1, it is indicated that the dilution is better than that of pure diesel. What is the reason for this effect?
|
||||||||||||||||||||||||||||||||||||||||||||
Response 1: Dear reviewer, the questions you raised are very crucial and are one of the starting points of this paper. Regarding the issue of increased viscosity, we consider the following potential causes: the viscosity of biodiesel is higher than that of diesel. During the experiments, when biodiesel is mixed with diesel, adding a small amount of biodiesel actually reduces the overall viscosity; however, after exceeding a certain threshold, the viscosity tends to increase. The reason may be that biodiesel possesses good lubricating properties, which are related to factors such as the unsaturation of fatty acid methyl esters (FAME) and carbon chain length. The ester groups contained in biodiesel are polar functional groups, which can more effectively dilute asphalt compared to the alkyl groups in low-sulfur diesel. However, when biodiesel is in excess, its increased concentration in the overall diluent leads to a further rise in viscosity. Therefore, a small amount of biodiesel acting in conjunction with diesel can better reduce viscosity, whereas an excess of biodiesel will instead increase viscosity. Currently there is little research on FTIR functional groups in this area, but we will continue to explore the relationship between functional groups and viscosity in this area, and hopefully in the future we will be able to find a link between them.
|
||||||||||||||||||||||||||||||||||||||||||||
Comments 2: Regarding the results shown in Figure 3, it is not clear. Could you provide more details?
|
||||||||||||||||||||||||||||||||||||||||||||
Response 2: Certainly, dear reviewer, this was our oversight. Figure 3 uses the AC-13 gradation, and the specific gradation information will be provided in the table below. We hope this information will help you understand the paper better and provide your suggestions. Please let us know if you need any more information on this subject.
Comments 3: Figure 4 quality needs to be improved, as it does not identify what is intended to be shown with these results.
Response 3: Certainly, dear reviewer, thank you for your suggestions. We will reorganize Figure 3 according to the standards of Figure 2 to ensure that the meaning of Figure 4 is clear. The main purpose of Figure 4 in this paper is to illustrate that different amounts of cold patch asphalt can lead to varying rates of aggregate coating. A small amount of cold patch asphalt can result in uneven coating of the aggregate, causing the phenomenon of “white material.” Conversely, an excess of cold patch asphalt can cause the aggregate to become “oily,” resulting in a decline in the performance of the mixture. This is the message that Figure 4 aims to convey.
Comments 4 : Regression coefficient is low, how can this result/trend be explained?
Response 4 : Dear reviewer, the low regression coefficients in this paper may be attributed to several reasons: In regression analysis, a low regression coefficient may arise from various factors. Firstly, the growth trend of asphalt mixtures is nonlinear, and their strength variations are difficult to predict, often resulting in significant dispersion, which is a normal phenomenon. Secondly, high variability in the data can increase the dispersion around the regression line, thereby reducing the model's R-squared value, even when the regression coefficients are statistically significant. The issue of multicollinearity, where the explanatory variables in the model are highly correlated, can lead to unstable estimates of regression coefficients, thereby affecting their significance. Additionally, issues related to data quality, such as inaccurate data sources or computational errors, may also lead to an underestimation of the regression coefficients. Finally, complex nonlinear relationships or endogeneity issues, where the explanatory variables are correlated with the error term, may also lead to biased estimates of the regression coefficients. Therefore, when regression coefficients are low, it is necessary to consider and analyze these aspects comprehensively to ensure the accuracy and reliability of the model results. Finally, The regression coefficient in this paper is approximately 0.92, which is considered accurate relative to other studies (refer to the article "Prediction and Optimization of Asphalt Mixtures Performance Containing Reclaimed Asphalt Pavement Materials and Warm Mix Agents Using Response Surface Methodology"). Therefore, the regression coefficient in this paper meets the required standards and is acceptable.
|

Round 2
Reviewer 3 Report
Comments and Suggestions for Authors
The authors responded to the various comments made in the first review. The discussion on relevant points was improved, some results were clarified and the quality of figures that were of low quality was improved.